# Mid-Infrared 2.79 μm Band Er, Cr: $Y_3Sc_2Ga_3O_{12}$ Laser Transmission Anti-Bending Low-Loss Anti-Resonant Hollow-Core Fiber

**Lei Huang** [1,2], **Peng Wang** [1], **Yinze Wang** [1,2], **Tingqing Cheng** [2], **Li Wang** [3] **and Haihe Jiang** [1,2,*]

1    Anhui Province Key Laboratory of Medical Physics and Technology, Institute of Health and Medical Technology, Hefei Institutes of Physical Science, Chinese Academy of Sciences, Hefei 230031, China; hl930523@mail.ustc.edu.cn (L.H.)

2    University of Science and Technology of China, Hefei 230026, China

3    Anhui Institute of Optics and Fine Mechanics, Hefei Institutes of Physical Science, Chinese Academy of Sciences, Hefei 230031, China

\*    Correspondence: hjiang@hfcas.ac.cn

**Abstract:** A large core size and bending resistance are very important properties of mid-infrared energy transfer fibers, but large core sizes usually lead to the deterioration of bending properties. A negative-curvature nested node-free anti-resonant hollow-core fiber (AR-HCF) based on quartz is proposed. It was made by adding a nested layer to a previous AR-HCF design to provide an additional anti-resonance region while keeping the gap between adjacent tubes strictly correlated with the core diameter to produce a node-free structure. These features improve the fiber's bending resistance while achieving a larger core diameter. The simulation results show that the radial air–glass anti-resonant layer is increased by the introduction of the nested anti-resonant tube, and the weak interference overlap between the fiber core and the cladding mode is reduced, so the fiber core's limiting loss and sensitivity to bending are effectively reduced. When the capillary wall thickness t of the fiber is 0.71 μm, the core diameter D is 70 μm, the ratio of the inner diameter of the cladding capillary to the core diameter d/D is 0.62, the diameter of the nested tube is $d_0$ = 29 μm, the fiber has a lower limiting loss at the wavelength of 2.79 μm, and the limiting loss is $3.28 \times 10^{-4}$ dB/m. At the same time, the optimized structure also has good bending resistance. When the bending radius is 30 mm, the bending loss is only $4.72 \times 10^{-2}$ dB/m. An anti-bending low-loss micro-structure hollow fiber with a bending radius of less than 30 mm was successfully achieved in the 2.79 μm band. An anti-bending low-loss anti-resonant hollow-core fiber with this structure constitutes a reliable choice for the light guiding system of a 2.79 μm band Er, Cr: YSGG laser therapy instrument.

**Keywords:** laser delivery; Er, Cr: YSGG; anti-resonant hollow-core fiber; bending loss

## 1. Introduction

Mid-infrared 2.79 μm band lasers are located in the strong absorption peak region of water molecules and hydroxyapatite, and they have important application value and potential in biomedicine, remote sensing, military equipment, and other fields due to their special wavelength [1]. In the field of biomedicine, carriers for flexible laser transmission (i.e., transmission along paths of arbitrary shape) are the focus of research on 2.79 μm medical laser devices. The traditional 2.79 μm Er, Cr: YSGG medical solid-state laser uses a light guide arm composed of multi-reflection lenses to transmit the laser beam. The transmission loss is large because of the lenses at the multiple joints of the laser beam path. Moreover, the rigid structure makes it difficult to realize the flexible operation of the optical carrier in any direction under the small bending radii required for use in the mouth and nasal cavity. Therefore, the light guide arm has severe limitations as a carrier for transmitting 2.79 μm band lasers. The use of fiber to transmit 2.79 μm band lasers not only

ensures flexible transmission but also avoids the interference of the external environment, thereby improving the transmission efficiency, and the better bending characteristics of fiber can also provide safer protection for medical services.

Examples of solid-core fibers that can transmit mid-infrared 2.79 μm band lasers include $SiO_2$ fibers, ZBLAN fibers [2], and $Al_2O_3$ fibers [3,4]. However, previously used fibers have shown significant faults when used for this purpose. Quartz fiber have large transmission loss, and ZBLAN fibers have poor bending performance. Also, sapphire fiber is expensive and short production lengths of it cannot achieve single-mode transmission. Solid-core fiber has limitations in mid-infrared 2.79 μm band laser transmission. However, Pryamikov et al. [5] adopted anti-resonant hollow-core fiber to achieve the low-loss flexible transmission of a laser with a wavelength longer than 3.5 μm, so the intrinsic problem of solid-core fiber transmission in the mid-infrared band was successfully solved. This kind of fiber has now become a research hotspot in the application of mid-infrared laser transmission. This anti-resonant hollow-core fiber is composed of a single-layer quartz tube, which is a structure that is simple and easy to pull. Mainly due to the anti-resonance guide light and the inhibition of the coupling effect, this fiber can achieve small loss and large width [6] at the same time because the glass material occupies just a small proportion of the fiber, and the laser propagates in an air core similar to in a free space environment [7]. This fiber has the characteristics of a high damage threshold and extremely low transmission loss [8–12].

Because bending loss is a vital consideration for hollow-core fibers in practical applications, this aspect is particularly worth studying. Research progress on bending loss in the near-infrared band has been relatively successful. In 2022, Zhao et al. [13] designed a nested hollow-core fiber with a bending loss of 0.001 dB/m at a minimum bending radius of 50 mm and a wavelength of 1.55 μm. In the same year, Liu et al. [14] designed an AR-HCF with a glass sheet nested structure and calculated that its bending loss of $LP_{01}$–$LP_{31}$ mode was less than $3.0 \times 10^{-4}$ dB/m in the wavelength range of 1.4–1.61 μm when the bending radius was 60 mm. However, research progress on bending loss in the mid-infrared band has been relatively slow, mainly because the wavelength is longer and it is more sensitive to the loss caused by bending deformation. In 2012, Urich et al. [15] used a triangular cladding AR-HCF to transmit a 2.94 μm high-energy microsecond pulse laser and measured a bending loss of 0.183 dB/m $\pm$ 0.05 dB/m at a minimum bending radius of 250 mm. In 2012, Fei Yu et al. [16] from the University of Bath in the United Kingdom designed an ice-cream AR-HCF, which achieved a transmission loss of 0.034 dB/m at a wavelength of 3.05 μm, and its light intensity transmitted at a bending radius of 200 mm was less than half that at a bending radius of 600 mm. In 2014, Knight et al. [17] of the University of Bath in the United Kingdom adopted an AR-HCF composed of 10 capillaries at a wavelength of 3.1 μm to achieve 0.1 dB/m transmission loss, with a bending transmission loss of 0.3 dB/m at its minimum bending radius of 80 mm. The above research on the transmission of infrared lasers using hollow-core fibers with different single-cladding structures achieved low transmission loss, but unfortunately, the single-cladding structure deviates from the desired matching of the core diameter, and the cladding capillary gap during bending deformation is severe, resulting in a large minimum bending radius and high bending loss.

In recent years, the anti-bending properties of nested anti-resonant hollow-core fibers have attracted much attention. In 2019, Mariusz Klimczak et al. [18] of the University of Warsaw reported a nested anti-resonant hollow-core fiber with a core diameter of 62 μm that can achieve mid-infrared band transmission, with a bending loss of 0.5 dB/m at a wavelength of 4 μm and a bending radius of 40 mm. When the bending radius is 15 mm, the bending loss is 5 dB/m, showing that this nested anti-resonant hollow-core fiber has good bending resistance in the mid-infrared band. AR-HCF is the ideal fiber for laser transmission in the 2.79 μm band. However, a medical laser transmission system usually requires a larger core diameter in order to transmit more laser energy, and a large core size will lead to a deterioration in bending performance. Therefore, a large core size to obtain a large mode field area and strong bending resistance with low bending loss are very

important performance indicators of mid-infrared fiber. To some extent, a fiber's ability to mechanically and optically bend to a relatively small radius determines whether it can enter commercial and practical applications.

In this paper, a negative-curvature nested anti-resonance hollow-core fiber based on quartz is designed, which provides additional anti-resonance region by adding a nesting layer, keeps the gap between adjacent tubes and the core diameter consistent, and makes each capillary have no nodes; this combination improves the fiber's bending resistance. Moreover, it has the characteristics of a large core diameter, simple structure, high flexibility, and simple manufacturing process. Finite element and control variable methods are used to study the transmission performance of the structures with different parameters. The structural parameters affecting the bending loss are optimized, which significantly improves the transmission characteristics of the optimized anti-resonant hollow-core fiber. The introduction of the nested anti-resonant tube increases the radial air–glass anti-resonant layer, and the embedded tube can reduce the weak interference overlap between the fiber core and the cladding mode, effectively reducing the limiting loss and bending loss of the fiber core. The simulation proves that it is feasible to transmit mid-infrared medical 2.79 μm band lasers, so this new fiber is expected to replace the traditional guide arm system currently used in 2.79 μm medical lasers.

## 2. AR-HCF Is Designed to Transmit 2.79 μm Band Lasers

Figure 1 shows the radial section structure of the new fiber. AR-HCF is composed of six tubes surrounded by capillary walls with the thickness t. The core is the largest tangent circle with the diameter D, while the inner diameter of each capillary is d, and the gap between capillaries is g. The white part is air, and the brown part is quartz, the base material. The six non-contacting capillaries form a cladding structure. The non-contacting geometry can eliminate Fano resonance [19], so each tube is regarded as an isolated tube and is tangent to the outer layer, which mainly plays roles of protection and support.

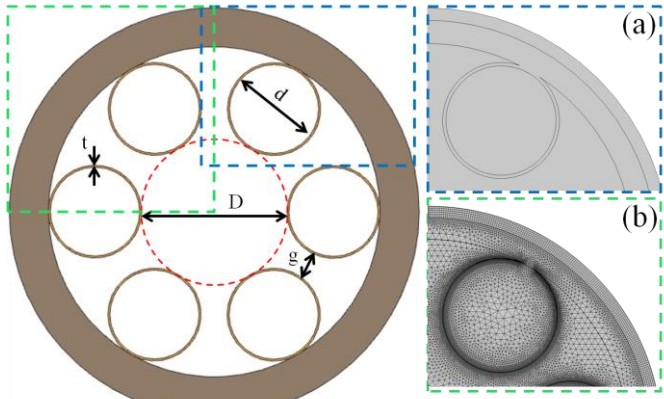

**Figure 1.** Cross-sectional structure diagram of anti-resonant hollow-core fiber. (**a**) Post-processing structure diagram at the intersection of the capillary and the outer layer. (**b**) Grid division diagram of a fiber structure quarter.

The fiber characteristics were calculated using numerical methods, namely finite element mode solvers (COMSOL Multiphysics), along with optimized mesh sizes and perfectly matched layers (PMLs). In the numerical simulation, the outermost layer of the outer envelope was set as the perfect matching layer, which is the critical edge that only absorbs and does not reflect. The intersection of the capillary and the outer layer was optimized (Figure 1a), greatly improving the accuracy of the calculation. The perfectly matched layer grid is a map with a maximum cell size of λ/6, and the rest of the grid is divided into a free triangle grid (Figure 1b). The maximum cell size distribution in the air and quartz regions is λ/5 and λ/4, respectively. The refractive index of air is

1, and the refractive index of quartz is related to the wavelength ($\lambda$). According to the Sellmeier [20] equation, the refractive index at 2.79 μm is 1.424. Figure 2 shows the mode field distribution of $LP_{01}$ (Figure 2a) and $LP_{11}$ (Figure 2b) of the anti-resonant hollow-core fiber at a wavelength of 2.79 μm. It can be seen that the electric field is confined well within the air fiber core.

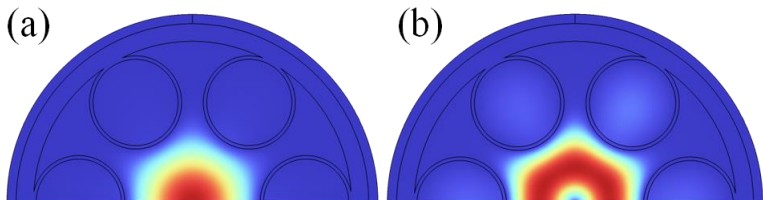

**Figure 2.** Mode field distribution of anti-resonant hollow-core fiber at wavelength of 2.79 μm. (**a**) Fundamental mode; (**b**) High-order mode.

## 3. Parameter Optimization

### 3.1. Effect of Capillary Wall Thickness on AR-HCF Transmission Characteristics

For anti-resonant hollow-core fibers, the capillary wall thickness is an important parameter that directly affects the transmission characteristics of the fiber. When the fiber material and its refractive index are determined, the capillary wall thickness directly determines the resonance and anti-resonance wavelength. When the capillary wall thickness $t$ satisfies Equation (1), then the wavelength of 2.79 μm is in the anti-resonant wavelength region, and the light in the fiber that does not resonate with the cladding formed by the capillary wall is reflected back into the core and transmitted in the core with low loss. When $t$ satisfies Equation (2), the 2.79 μm wavelength is in a resonant state and the light in the air fiber core will not be restricted, and at the same time, it will leak into the capillary cladding, resulting in a sharp increase in fiber loss and even damage to the cladding structure [21].

$$t_1 = \frac{(m - 0.5)\lambda}{2\sqrt{n_1^2 - n_0^2}} \tag{1}$$

$$t_2 = \frac{m\lambda}{2\sqrt{n_1^2 - n_0^2}} \tag{2}$$

Here, $\lambda$ is the wavelength, $m$ is an integer, $n_0$ is the air with a refractive index of 1, $n_1$ is the quartz refractive index (1.424 at the wavelength of 2.79 μm), $t_1$ is the thickness of the capillary wall at the time of anti-resonance, and $t_2$ is the thickness of the capillary wall at the time of resonance.

The initial conditions were set as follows: the core diameter D was 30 times the wavelength 83.7 μm, and the ratio d/D of the capillary inner diameter to the core diameter was kept unchanged at 0.68. The parametric scanning interval of $t$ was 0.3–2.0 μm. The step was set to 0.01 μm. The parameters are shown in Table 1. Figure 3a shows the anti-resonance curve of constraint loss with cladding capillary wall thickness when the anti-resonance wavelength was 2.79 μm, showing the periodicity of alternating anti-resonance and resonance action. The capillary wall thickness does not need to be completely consistent with the wall thickness calculated by Equation (1), and an anti-resonance effect can also be generated near it. In the resonant region (1.33 μm < $t$ < 1.58 μm), the core mode is partially coupled with the cladding mode, which leads to a large confinement loss of several hundred dB/m. As can be seen from the illustration, a large number of electric fields are distributed in the capillary cladding. In the anti-resonance region (t > 1.58 μm, $t$ < 1.33 μm), the core mode is well confined in the fiber core, and the constraint loss is low. Figure 3b shows the variation curve of the effective refractive index of the core with the capillary wall thickness. In the resonant region of 1.33 μm < $t$ < 1.58 μm, the effective refractive index is not continuous, and the resonance between the light and the quartz

cladding is the largest, which explains the root cause of the high loss in Figure 3a. The coupling of the core mode and the cladding mode causes the discontinuity of the effective refractive index of the fundamental mode.

**Table 1.** Range of capillary wall thickness parameters and other fixed parameters.

| Name of Parameter | Value |
|---|---|
| Core diameter | D = 83.7 μm |
| Capillary wall thickness | 0.3 μm < t < 2 μm |
| Ratio of capillary inner diameter to core diameter | d/D = 0.68 |

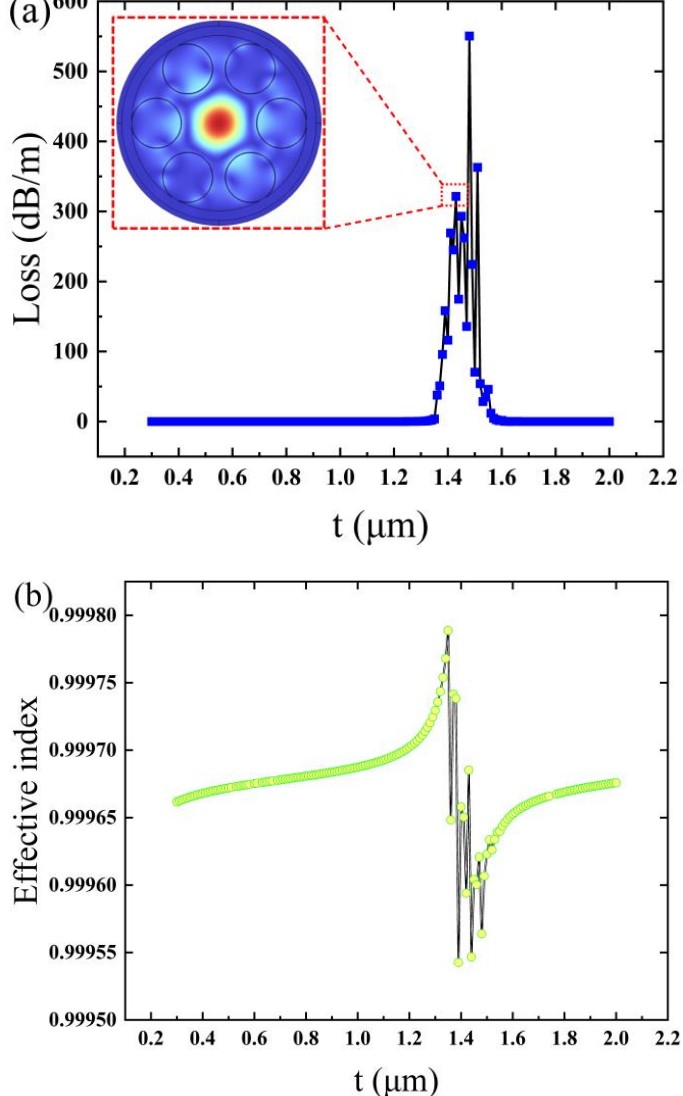

**Figure 3.** (**a**) Transmission loss and (**b**) effective refractive index changes with capillary wall thickness t.

To find the optimal capillary wall thickness, the anti-resonance region (0.3 μm < t < 1.2 μm) was scanned with a step length of 0.001 μm, as shown in Figure 4a. At t = 0.723 μm, the basic mode loss of $LP_{01}$ reached its minimum at $8 \times 10^{-3}$ dB/m. For anti-resonant hollow-core fibers, the single-mode characteristics are usually described by the high-order mode extinction ratio (*HOMER*), which is calculated by comparing the minimum $LP_{11}$ mode loss with the core $LP_{01}$ mode loss [22]:

$$HOMER = \frac{loss_{LP_{11}}}{loss_{LP_{01}}} \tag{3}$$

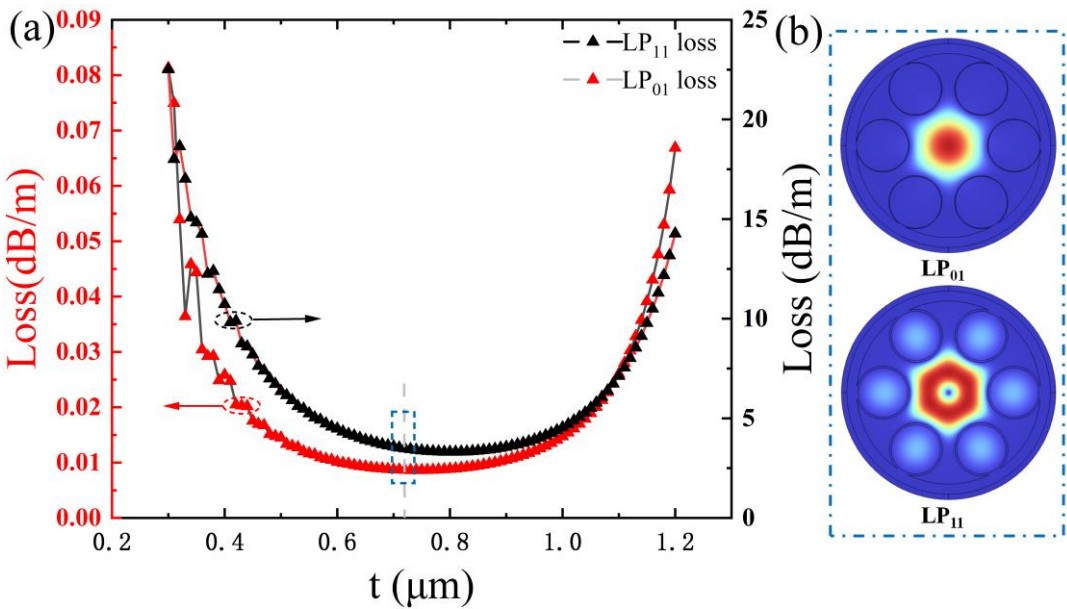

**Figure 4.** (**a**) The change in limiting loss in the range of *t* values 0.3–1.2 μm, and (**b**) the mode field distribution of core fundamental mode and high-order mode when *t* = 0.723 μm.

When the *HOMER* value is greater than 100, the fiber can be considered to maintain single-mode transmission. At *t* = 0.723 μm, the high-order mode extinction ratio reaches 414. Figure 4b shows the mode field distribution of $LP_{01}$ and $LP_{11}$ in the fiber core at *t* = 0.723 μm. It can be seen that the mode of $LP_{01}$ is well confined in the air fiber core, and no electric field distribution occurs in the capillary wall cladding. The $LP_{11}$ mode is distributed in the air core and cladding tube, resulting in increased loss, which is advantageous for single-mode transmission fibers.

When the anti-resonant hollow-core fiber is used as the light guide system of a 2.79 μm laser medical instrument, bending is inevitable in clinical applications and will introduce bending loss. When the fiber is bent, the refractive index of the cladding layer and the core changes, which changes the phase of the core mode and cladding mode. When the phase between the core mode and cladding mode meets certain matching conditions, the core mode leaks into the cladding, resulting in bending loss. With a reduction in the bending radius, the bending loss gradually becomes the dominant optical fiber transmission loss and even affects the transmission mode of the optical fiber to produce disturbance modes, which lose their constraints in the core, resulting in the bending loss. Therefore, to determine how to preserve the fiber's ability to maintain high-efficiency and low-loss single-mode transmission under different bending radii, we studied the losses corresponding to different capillary wall thicknesses and bending radii to find the most suitable capillary wall thicknesses, as shown in Figure 5. In this numerical calculation, the core effective refractive index ($n_{eql}$) and cladding refractive index ($n_{eql\_silica}$) of the bent fiber can be changed based on the straight fiber's characteristics:

$$n_{eql} = n_l . \exp(x/R_b) \tag{4}$$

$$n_{eql\_silica} = n_{silica} . \exp(x/R_b) \tag{5}$$

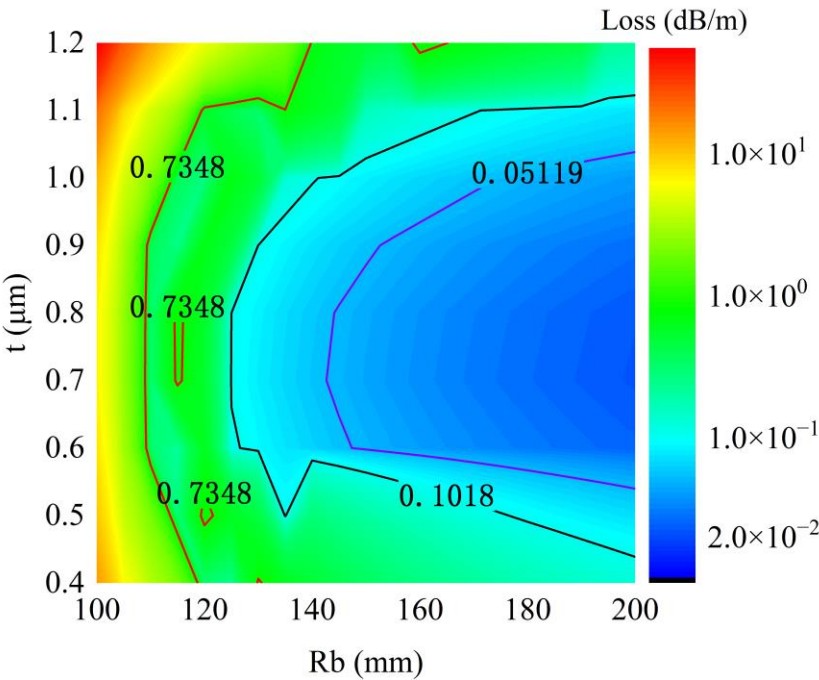

**Figure 5.** Surface diagram of bending loss table for adjusting different capillary wall thicknesses.

Here, $R_b$ is the bending radius, $n_l$ is the core refractive index of the straight fiber, $n_{silica}$ is the cladding refractive index of the straight fiber, and $x$ is the transverse distance from the center of the curved fiber.

The combined parameterization scans the loss of the anti-resonance region of the capillary wall thickness (0.4 μm < t < 1.2 μm) with bending radii of 100 to 200 mm. Figure 5 shows that when $t = 1.2$ μm, the bending loss is sensitive to the bending radius, and when the bending radius is 100 mm, the bending loss reaches 53.4 dB/m, although the bending loss can also reach 1 dB/m under a larger bending radius. When $t = 0.71$ μm, the bending loss is not sensitive to the bending radius. When the bending radius is 115 mm, the bending loss is 0.7348 dB/m, and when the bending radius is 150 mm, the bending loss is only 0.05119 dB/m. As the result of this testing, $t = 0.71$ μm was selected to achieve AR-HCF with practical resistance to loss due to bending.

### 3.2. Influence of Capillary Inner Diameter on AR-HCF Transmission Characteristics

Changes in capillary inner diameter directly affect changes in core diameter and the loss of core basic mode and high-order mode, so optimizing capillary inner diameter is also very important. Based on the parameter optimization in Section 3.1, the core diameter $D = 83.7$ μm and capillary wall thickness $t = 0.71$ μm remained unchanged, and d/D varied within a scanning range of 0.4–0.9. The results are shown in Figure 6. Changing the inner diameter of the cladding capillary also changes the capillary gap g, and this inter-tube gap is also an important part of the cladding effect, so its influence on the optical properties should also be quantified. The relationship of each geometry can be expressed as:

$$g = (D + d)\sin(\pi/N) - d \tag{6}$$

where N represents the number of cladding tubes. Figure 6 shows that with the increase in the inner diameter of the capillary cladding, the capillary gap decreases linearly, and the loss decreases. When d/D = 0.64 and the inner diameter of the cladding tube d = 53.6 μm, the loss reaches the minimum value of $8.4 \times 10^{-3}$ dB/m. When d/D < 0.64, the loss increases significantly as d decreases because a decreasing d corresponds to an increasing tube spacing, which will cause light leakage [23–26]; see Figure 7b. Another reason we consider is that with decreasing d, the total perimeter of the capillary cladding decreases,

and the spatial overlap increases between the modes of the air layer and the modes of the glass layer; that is, the coupling strength increases, which leads to the loss of the modes of the air layer. When d/D > 0.64, the loss increases rather than decreases. The main reason is that when d increases past a certain threshold, the surface tension between the cladding capillaries leads to the formation of local nodes between the capillaries, which greatly increases the fiber's transmission loss. The transmission loss is consistent with the change in the imaginary part of the effective refractive index (Figure 8).

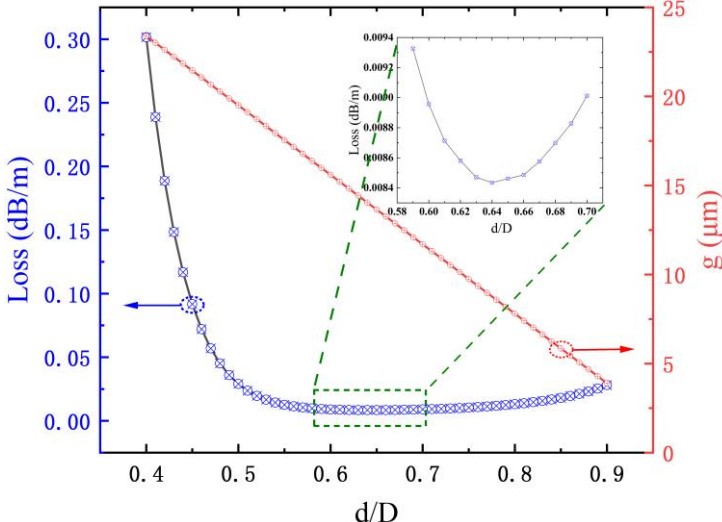

**Figure 6.** Variation curve of transmission loss of anti-resonant hollow-core fiber with d/D.

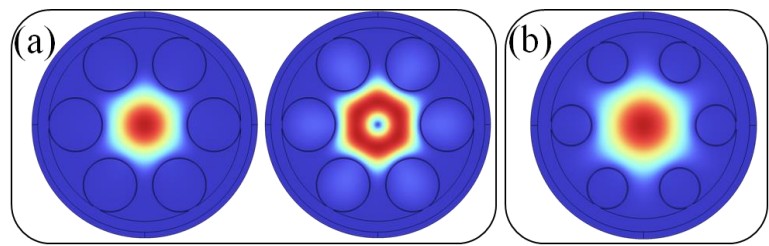

**Figure 7.** (**a**) d/D = 0.64 core pattern field distribution, (**b**) d/D = 0.4 core pattern distribution.

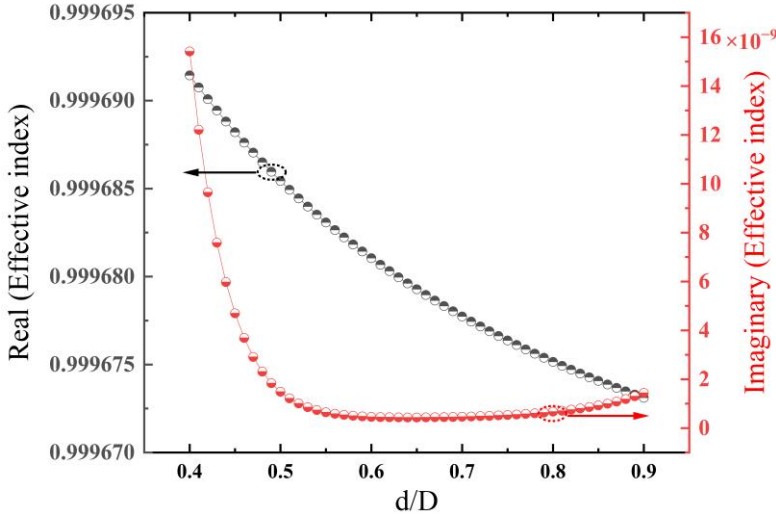

**Figure 8.** The relationship between the imaginary and real parts of the effective refractive index with d/D.

We used the same combination of parameters to scan for the effect of the d/D ratio on bending losses. For this testing, we used a d/D value with a small loss (0.6–0.8) and a bending radius of 100–200 mm, representing a very limiting bending radius range for most anti-resonant air-core fibers. The parameters are shown in Table 2. As can be seen from Figure 9, when d/D < 0.62, the fiber is not sensitive to bending radii as low as 100 mm, where, as can be seen from the red contour line in the figure, the bending loss is only 0.7418 dB/m. The choice of d/D = 0.62 ensures low loss and low sensitivity to bending radius.

**Table 2.** Parameter range of the ratio d/D of capillary inner diameter to fiber core diameter and other fixed parameters.

| Name of Parameter | Value |
| --- | --- |
| Core diameter | D = 83.7 μm |
| Capillary wall thickness | t = 0.71 μm |
| Ratio of capillary inner diameter to core diameter | 0.4 < d/D < 0.9 |

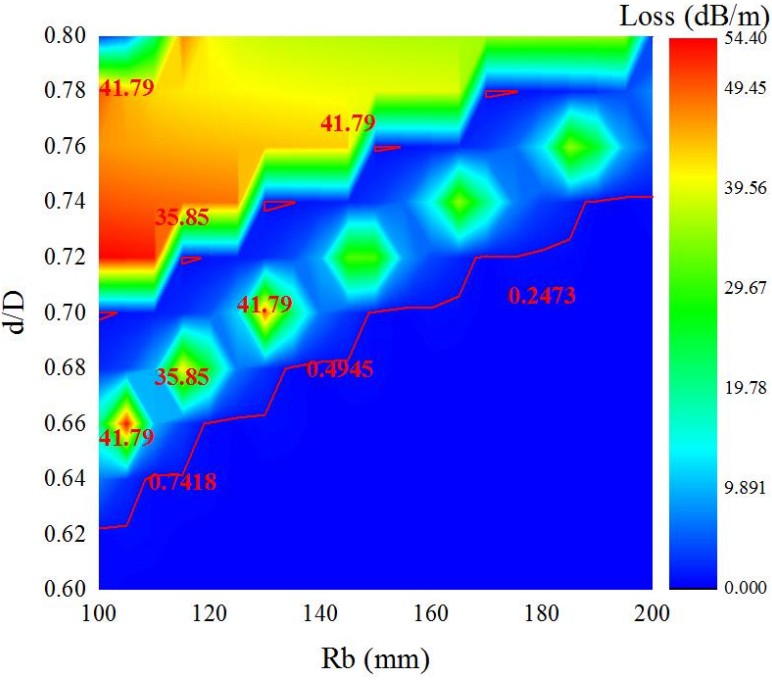

**Figure 9.** Effect of d/D ratio on fiber loss at different bending radii.

### 3.3. Influence of Fiber Core Diameter on AR-HCF Transmission Characteristics

The core diameter directly affects AR-HCF loss. The optimal core diameter can be obtained by numerical calculation based on the relationship between fiber loss and core diameter. After fixing d/D = 0.62 and *t* = 0.71 μm, the relationship between loss and core diameter can be obtained by parametric scanning of D, as shown in Figure 10. It can be seen from the figure that the fiber loss decreases with the increase in the core diameter. When D = 120 μm, the loss is the lowest, reaching $3.0 \times 10^{-3}$ dB/m, and the high-order mode extinction ratio reaches 935. Without considering the bending loss, D = 120 μm is the best choice. Figure 11 shows the relationship between the real and imaginary parts of the effective refractive index of the fiber core and the diameter of the fiber core. The real part increases with increasing diameter. The main reason is that the larger the diameter of the core, the farther the center of the fiber transmission is from the cladding tube wall, so the lower the coupling effect between the fiber core transmission mode and the cladding tube mode, and the closer the real part of the effective refractive index of the fiber core fundamental mode is to the air refractive index.

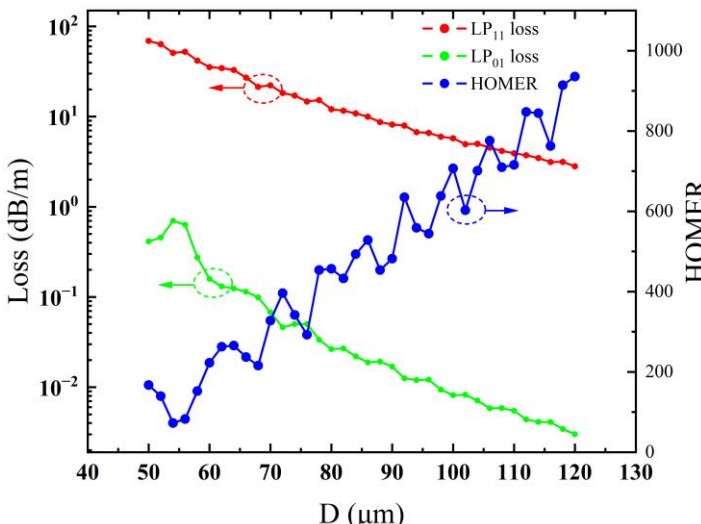

**Figure 10.** Relationship between core base mode and high-order mode loss, high-order mode extinction ratio, and core diameter.

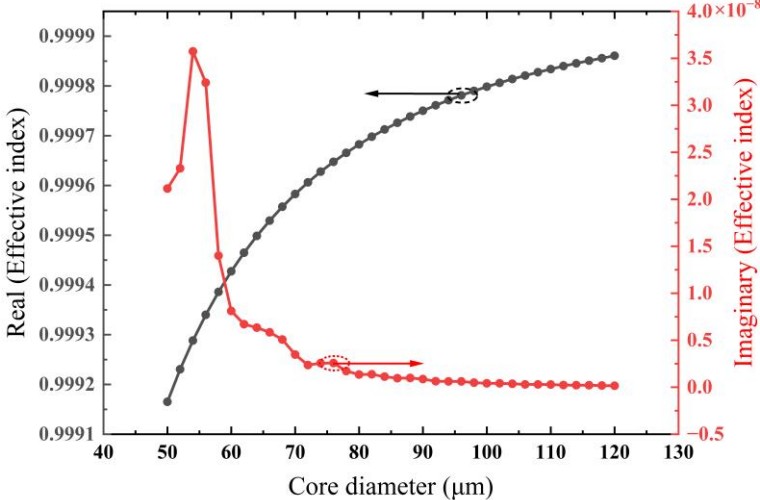

**Figure 11.** Relationship between the real and imaginary parts of the core effective refractive index and the core diameter.

To obtain the anti-bending characteristics of the new AR-HCF, we varied the loss of D in the lower loss range of 70–90 μm corresponding to different bending radii (100–200 mm) to find the optimal D value; Figure 12 shows the results. The parameters are shown in Table 3. When the bending radius R = 100 mm, the bending loss at D = 70 μm is the lowest, which can be seen from the contour line to 0.3089 dB/m. When D > 80 μm, the bending loss is greater than 37 dB/m, which shows the AR-HCF's reat sensitivity to the bending radius. Figure 13 shows the distribution diagram of the core fundamental mode field corresponding to different D values when the bending radius R = 100 mm. It can be seen from the diagram that when D = 70 μm, the fundamental mode is confined well in the air core, and only a small part of the electric field leaks into the cladding tube, which corresponds to the low loss in Figure 12. When D = 74 μm, a large part of the electric field leaks into the cladding tube, and the loss reaches 37.07 dB/m. When D = 86 μm, there is strong coupling between the cladding tube and the air fiber core mode, and only a small part of the field exists in the air core. In this situation, most of the field is coupled to the cladding tube, resulting in a huge increase in the loss value of 47.63 dB/m. The core electric field distribution diagram (Figure 13) corresponds to the loss diagram (Figure 12). Based

on these results, D = 70 μm was chosen as the best air core diameter for these anti-bending low-loss anti-resonant hollow-core fibers.

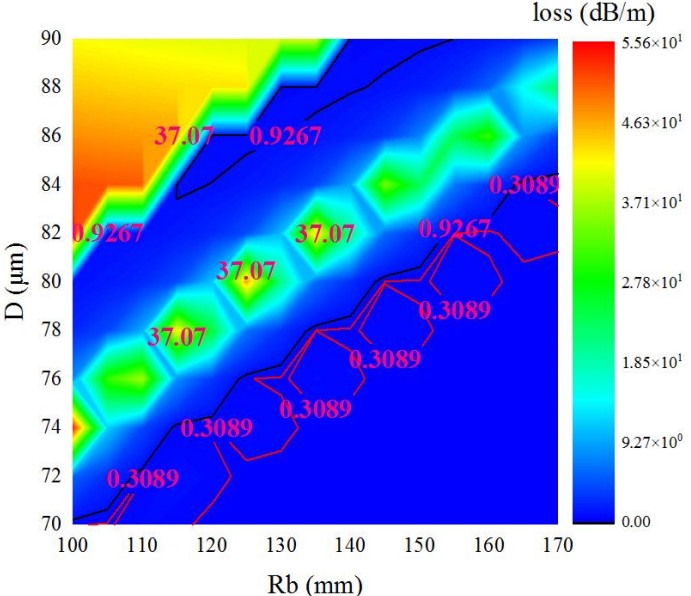

**Figure 12.** Fiber bending loss at different core diameters D and bending radii.

**Table 3.** Parameter range of core diameter D and other fixed parameters.

| Name of Parameter | Value |
|---|---|
| Core diameter | 50 μm < D <120 μm |
| Capillary wall thickness | t = 0.71 μm |
| Ratio of capillary inner diameter to core diameter | d/D = 0.62 |

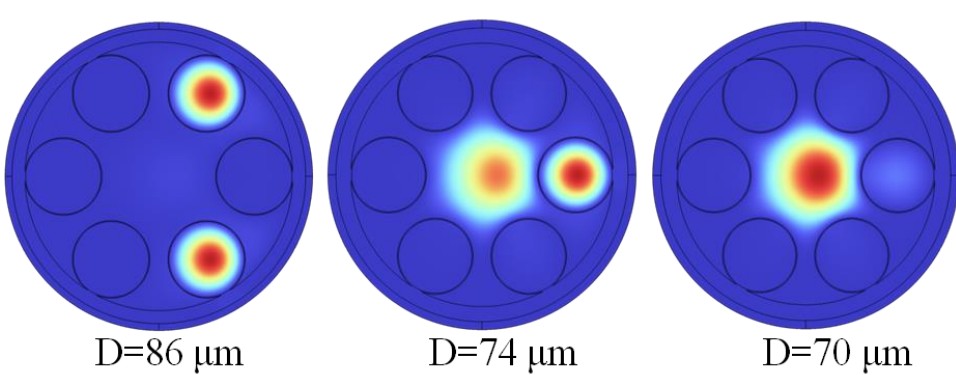

**Figure 13.** When the bending radius is 100 mm, the electric field distribution of D = 86 μm, 74 μm, and 70 μm core fundamental mode.

## 4. Results and Discussion

### 4.1. Influence of Inner Diameter of Casing on Bending Loss

Section 3 analyzed how different capillary wall thicknesses, ratios of cladding wall thickness to core, and core diameters affect the transmission characteristics. The analysis revealed that the transmission loss is 0.02 dB/m when t = 0.71 μm, d/D = 0.62, D = 70 μm, and the bending loss is 0.31 dB/m when the bending radius R = 100 mm. To further reduce the fiber's transmission loss and its sensitivity to the bending radius, we introduced a nested anti-resonant tube based on the above structural parameters in order to increase the

radial air–glass anti-resonant layer and reduce the limiting loss of the fiber core. We denote the wall thickness of the embedded capillary as $d_0$. The structure is shown in Figure 14.

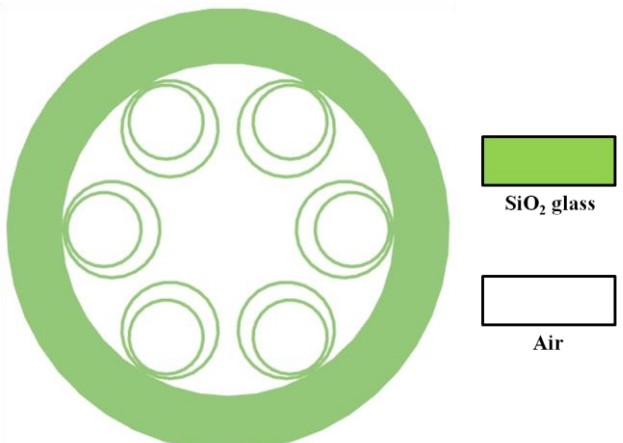

**Figure 14.** Structure diagram of nested anti-resonant hollow-core fiber.

Keeping the wall thickness of the embedded tube the same as the wall thickness of the outer capillary tube, the influence of different $d_0$ values on the loss in cases of small bending radius (10–100 mm) was studied; the results are shown in Figure 15. When the bending radius R = 30 mm and $d_0$ = 29 µm, the bending loss of the AR-HCF with a nested structure is $4.72 \times 10^{-2}$ dB/m. When R = 50 mm, the bending loss is $5.86 \times 10^{-3}$ dB/m, and when R = 100 mm, the bending loss is only $9.93 \times 10^{-4}$ dB/m. Compared with the single-layer structure, these results are three orders of magnitude lower (i.e., better) with a smaller bending radius and a larger bending degree. The fiber of this nested structure begins to be sensitive to the bending radius only when the bending radius is less than 30 mm. The righthand figure in Figure 15 shows the x and y polarization distribution of the core fundamental mode at the bending radius R = 30 mm and $d_0$ = 29 µm. It can be seen that the fundamental mode is well confined within the core. When a bending radius is fixed and the $d_0$ value is small, the casing gradually moves away from the core, which means that the casing cannot play a good role in confining the light of the core, resulting in a greater loss. When $d_0$ is large, the diameter of the casing becomes closer to the cladding capillary, which will cause the proximity area of the two glass rings to gradually become larger, thus forming a thick glass ring similar to the air defect, which will increase the fiber loss. Therefore, it is necessary to select an appropriate $d_0$ value to reduce the limiting loss and increase the bending resistance. In this paper, $d_0$ = 29 µm was selected according to the calculation results.

For comparison with single-layer structures, the bending losses for $d_0$ = 29 µm and R = 100 mm were calculated, as shown in Figure 16. In that illustration, we can see that the fundamental modes in both directions are well confined within the core, and there are no leaked patterns in the cladding. In contrast, in the single-cladding structure (Figure 13 (D = 70 µm)), we can see that some patterns leak into the cladding. The results show that adding a nesting layer can effectively inhibit the coupling between the cladding mode and fiber core mode to significantly reduce the bending loss. The curve of the virtual part of the effective refractive index changing with the bending radius in Figure 16b fundamentally explains that the nested structure (black curve) is not sensitive to the bending radius. The virtual part of the effective refractive index of the nested structure is always smaller than that of the single-cladding structure, and the virtual part of the effective refractive index can intuitively show the loss of the fiber. The bending loss of the single-cladding structure is obviously greater than that of the nested structure.

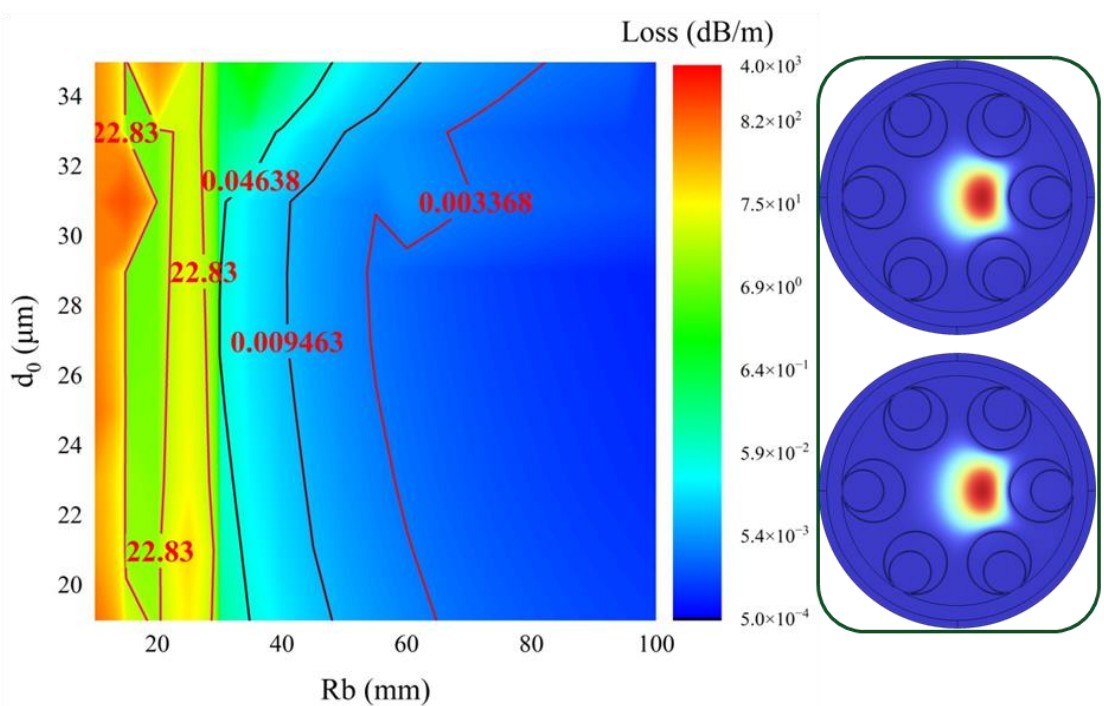

**Figure 15.** Fiber loss under different insert diameters and bending radii, and the distribution diagram of the core's fundamental mode field at R = 30 mm and $d_0$ = 29 μm.

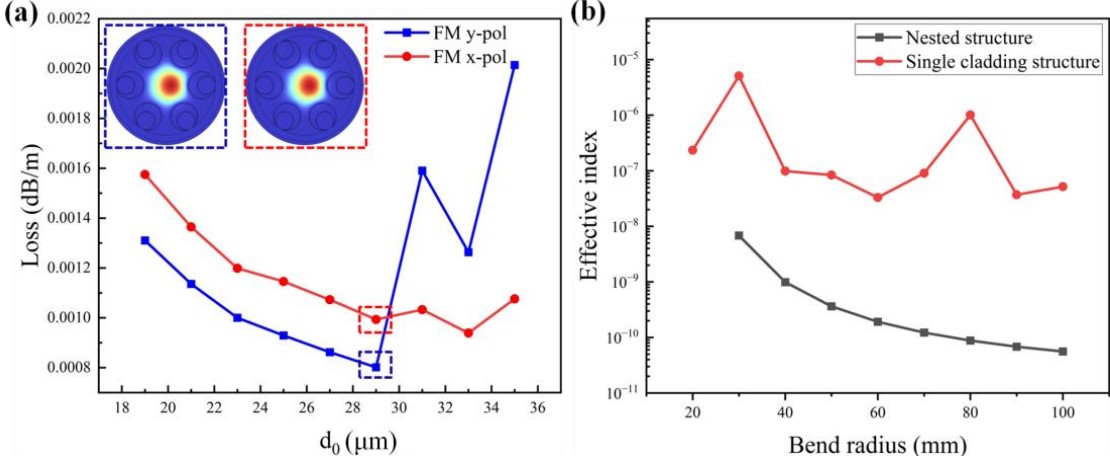

**Figure 16.** (**a**) The limiting loss value changes with $d_0$ when R = 100 mm. The illustration shows the x and y polarization fundamental mode field distribution of the fiber core when $d_0$ = 29 μm. (**b**) The variation curve of the imaginary part of the effective refractive index of the core of the nested structure and the single-cladding structure with the bending radius.

### 4.2. Influence of Capillary Gap on Bending Loss

The core diameter is 70 μm, the capillary wall thickness is 0.71 μm, the inner diameter of the embedded tube is 29 μm, and the wavelength used in the simulation is 2.79 μm. We now discuss the influence of the capillary gap on the bending loss of the nested fiber. At the bottom of Figure 17 is a larger view of some areas of the structure under consideration (the parts within the rectangles of different colors). It is worth noting that for all the simulated structures, in order to improve the accuracy of the simulation, a small penetration of 0.1 μm was added to all the cladding tubes, as illustrated by the enlarged diagram at the bottom of Figure 17b.

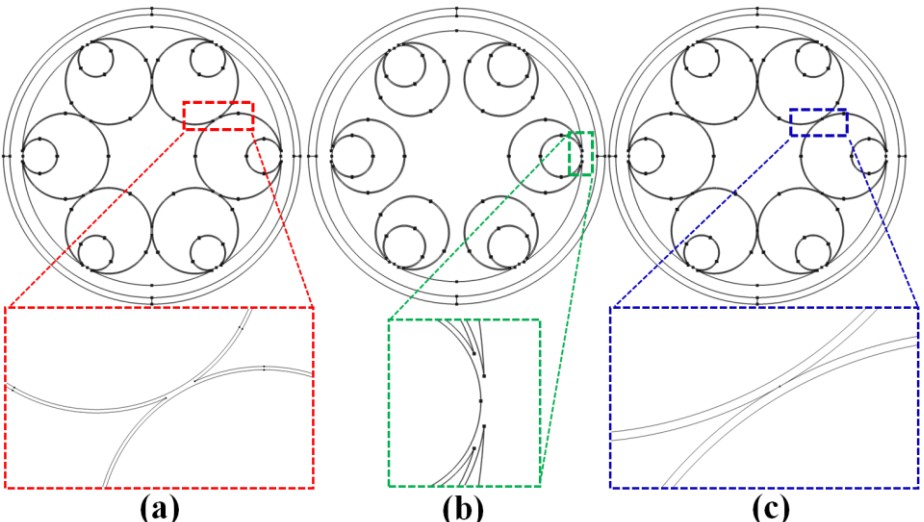

**Figure 17.** (**a**) Structure diagram of capillary gap less than 0, (**b**) structure diagram of capillary gap greater than 0, and (**c**) structure diagram of capillary gap equal to 0.

We changed the size of the gap by adjusting the ratio between the inner diameter of the outer layer tube and the diameter of the fiber core. For the convenience of expression, we defined this ratio as $\tau$, and the capillary gap was inversely proportional to $\tau$. The calculation results are shown in Figure 18. In the structure of $\tau = 1$ (yellow line in Figure 18, structure in Figure 17c), the capillary gap is 0, and each capillary is tangent. In this configuration, the bending loss is large due to the strong influence of the coupling between the core base mode and cladding mode, and the limiting loss increases due to the generation of nodes. It can also be noted that the bending loss decreases when the cladding tube spacing increases ($\tau$ decreases). For more suitable tube spacing, the anti-resonant capillary drives the electric field away from its surrounding area, which causes a small percentage of the power to leak through the capillary, so the loss is minimal. However, when the bending radius is greater than 40 mm, $\tau = 0.72$ corresponds to the smallest loss value. The main reason is that with the increase in pipe clearance, the leakage loss also increases. We conclude from this that since the amount of leakage increases as $\tau$ decreases, a correct balance is needed to design fibers with low bending losses.

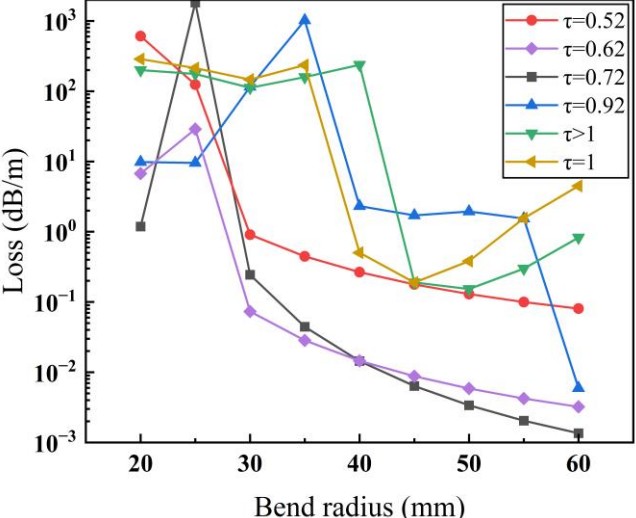

**Figure 18.** Bending loss corresponding to different capillary gaps.

## 5. Conclusions

In this paper, a negative-curvature nested anti-bending low-loss AR-HCF based on quartz material is proposed. The simulation results show that the radial air–glass anti-resonant layer is increased by introducing a nested anti-resonant tube, the weak interference overlap between the fiber core and cladding mode is reduced effectively, and the limiting loss of the fiber core is reduced effectively. The anti-resonant hollow-fiber structure has a low limiting loss of $3.28 \times 10^{-4}$ dB/m at the 2.79 μm band with good bending resistance since it can maintain a fundamental mode loss of less than $4.72 \times 10^{-2}$ dB/m at a bending radius as low as 30 mm. This bending radius is far better than the minimum bending radius of traditional solid-core fiber, allowing the new fiber to meet clinical requirements. We believe that the negative-curvature nested structure scheme and the designed anti-resonant hollow-core fiber with large core diameter open up new possibilities for application research on mid-infrared laser light conduction systems and the light transmission in 2.79 μm solid-state medical laser equipment.

**Author Contributions:** Conceptualization, L.H. and H.J.; methodology, L.H.; software, L.H. and P.W.; validation, Y.W.; formal analysis, T.C. and L.H.; investigation, L.H.; resources, L.H. and L.W.; data curation, L.H.; writing—original draft preparation, L.H.; writing—review and editing, L.H., L.W. and H.J.; visualization, L.H.; supervision, H.J.; project administration, T.C.; funding acquisition, T.C. and H.J. All authors have read and agreed to the published version of the manuscript.

**Funding:** This work was funded by the Wanjiang Center industrialization project (WJ21CYHXM06) and the National Key Research and Development Program of China (2018YFB0407204).

**Institutional Review Board Statement:** Not applicable.

**Informed Consent Statement:** Not applicable.

**Data Availability Statement:** Data underlying the results presented in this paper are not publicly available at this time but may be obtained from the authors upon reasonable request.

**Conflicts of Interest:** The authors declare no conflicts of interest.

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
