# Peer review of "Mid-Infrared 2.79 μm Band Er, Cr: Y3Sc2Ga3O12 Laser Transmission Anti-Bending Low-Loss Anti-Resonant Hollow-Core Fiber"

_photonics, doi:10.3390/photonics11050432_

Round 1

Reviewer 1 Report

Comments and Suggestions for Authors

Review paper: “Mid-infrared 2.79 μm band Er, Cr: YSGG laser transmission 2 anti-bending low-loss anti-resonant hollow-core fiber”

Author: Lei Huang, Peng Wang, Yinze Wang, Tingqing Cheng, Li Wang and Haihe Jiang

The paper presents theoretical analyses of AR- HCF from point of view of bending loss for transmitting wavelength 2,74μm. Mid-infrared 2.79 μm band laser is in the strong absorption peak region of water molecules and hydroxyapatite, and it has important application value and potential in biomedicine, remote sensing, military equipment, and other fields due to its special wavelength. The problem is in increasing of bending loss of AR-HCF for this wavelength region for medical application. This problem theoretically solved this paper.

The author suggested structure of AR-HCF and calculated loss as function of the bending diameter and different thick of capillary wall, its distance and hollow core diameter. The analyse results are parameters of AR-HCF with six capillaries, its diameter d and wall thick, core diameter D and distance between capillaries.

Theoretical result of loss for bend diameter of AR-HCF 30mm and wavelength 2,74μm was 3.28×10-4 dB/m at the 2.79 μm band and for fundamental mode loss of less than 4.72×10-2 dB/m.

I have following notes:

1.        The paper is very hard to read and oriented in properties of calculation, because the data for analyses and calculation are mentioned in the text. I recommended enter the data in a table for each suggest parameters of AR-HCF

2.        What was step of scanning interval of t (line 166)? It is important to know what accuracy must have be this parameter in practical realization of AR-HCF.

3.        It is not described N in Equation (6)

4.        Correct references (5), (7) – must be stated surname of authors.

5.        Can the authors explain practical preparation of their suggested AR-HCF. Requirements on its parameters must be in μm and it will be big problem from my point of view.

Result: On the notes base mention above, I can recommended this paper for publication after corrections are made.

Author Response

请参阅附件。

Reviewer 2 Report

Comments and Suggestions for Authors

In this paper, a negative curvature nested anti-bending low-loss anti-resonant hollow-fiber based on quartz material is proposed. The simulation results show that the radial air-glass anti-resonant layer is increased by introducing the nested anti-resonant tube, the weak interference overlap between the fiber core and cladding mode is reduced effectively, and the limiting loss of the fiber core is reduced effectively. Overall, the idea received my attention and the methodology is technically sound. However, there are some specific issues the authors should address by making modifications before we can proceed and positive action can be taken.

  1. Any claims of novelty make specific reference to what is known, unknown, accomplished and unaccomplished in scientific literature, rather than using words such as "new," "for the first time," "novel," or "best." The latter expressions may only be used if they can be fully substantiated, and, as such, should be avoided in the title, abstract, and conclusion.

  2. Please ensure all tables and figures have been numbered and cited in order in the text.

  3. Use the last name of the first author followed by “et al.” in your in-text citation.

  4. Any physical quantity which has dimensions must have a unit. i.e., d0 in figure 16.

  5. When you use an abbreviation in both the abstract and the text, define it in BOTH places upon first use. Further, after you define an abbreviation, use only the abbreviation. Do not alternate between spelling out the term and abbreviating it. i.e., YSGG, ZBLAN, …

  6. Please increase the font size in Figs. 3–7, to make sure that all micrographs have scale markers, and that all scale markers and text in figures are large enough to be easily read.

  7. The authors investigated the generation of lasers, have the authors noticed some papers on this point? i.e., [Independent degrees of freedom in two-dimensional materials. Phys. Rev. B 2020, 101, 081414(R), doi:10.1103/PhysRevB.101.081414.]…

Comments on the Quality of English Language

The English language requires improvements. Spelling and grammatical errors exist in the manuscript. i.e., l. 104: makes each capillary has…; l. 229: lager; l. 289: becomes to… We recommend you ask a native English speaker to edit the paper or use an independent professional editor.

Author Response

请参阅附件。

Reviewer 3 Report

Comments and Suggestions for Authors

The authors report through simulation the transmission of mid-infrared 2.79mm band laser considering a negative curvature nested hollow-core fiber based on quartz. The nesting layer improve the fiber bending resistance. Applying finite element and control variable methods it was studied the transmission characteristics considering the: effect of capillary wall thickness, fiber core diameter. With the aim to reduce the bending losses, it is introduced a nested tube. It is demonstrated the bending losses reduction.

In my opinion this paper is suitable for publication in Photonics. However, is necessary to review some issues:

-              On page 12, line 386.

a)        The figure that shown the calculation results is Figure 17.

b)        In Figure 18, the green line does not indicate the results for $tau$ = 1 as it is mentioned.

-              In Figure 18, green line presents the results for $tau$ < 1. Is it correct? In this figure, all the values before the green line are for $tau$ < 1.
